# Design of Phage-Cocktail–Containing Hydrogel for the Treatment of *Pseudomonas aeruginosa*–Infected Wounds

**DOI:** 10.3390/v15030803

**Published:** 2023-03-21

**Authors:** Fatemeh Shafigh Kheljan, Farzam Sheikhzadeh Hesari, Mohammad Sadegh Aminifazl, Mikael Skurnik, Sophia Goladze, Gholamreza Zarrini

**Affiliations:** 1Department of Animal Biology, Faculty of Natural Sciences, University of Tabriz, Tabriz 5166616471, Iran; fatemehshafigh20@gmail.com (F.S.K.); drhesari@gmail.com (F.S.H.); 2Department of Applied Chemistry, Faculty of Chemistry, University of Tabriz, Tabriz 5166616471, Iran; sadegamini@yahoo.com; 3Human Microbiome Research Program, Department of Bacteriology and Immunology, Faculty of Medicine, University of Helsinki and Helsinki University Hospital, 00014 HUS Helsinki, Finland; mikael.skurnik@helsinki.fi (M.S.); sophie.gholadze@helsinki.fi (S.G.)

**Keywords:** phage therapy, nosocomial infections, *Pseudomonas aeruginosa*, hydrogel, ciprofloxacin

## Abstract

Recently, the treatment of infected wounds has become a global problem due to increased antibiotic resistance in bacteria. The Gram-negative opportunistic pathogen *Pseudomonas aeruginosa* is often present in chronic skin infections, and it has become a threat to public health as it is increasingly multidrug resistant. Due to this, new measures to enable treatment of infections are necessary. Treatment of bacterial infections with bacteriophages, known as phage therapy, has been in use for a century, and has potential with its antimicrobial effect. The main purpose of this study was to create a phage-containing wound dressing with the ability to prevent bacterial infection and rapid wound healing without side effects. Several phages against *P. aeruginosa* were isolated from wastewater, and two polyvalent phages were used to prepare a phage cocktail. The phage cocktail was loaded in a hydrogel composed of polymers of sodium alginate (SA) and carboxymethyl cellulose (CMC). To compare the antimicrobial effects, hydrogels containing phages, ciprofloxacin, or phages plus ciprofloxacin were produced, and hydrogels without either. The antimicrobial effect of these hydrogels was investigated in vitro and in vivo using an experimental mouse wound infection model. The wound-healing process in different mouse groups showed that phage-containing hydrogels and antibiotic-containing hydrogels have almost the same antimicrobial effect. However, in terms of wound healing and pathological process, the phage-containing hydrogels performed better than the antibiotic alone. The best performance was achieved with the phage–antibiotic hydrogel, indicating a synergistic effect between the phage cocktail and the antibiotic. In conclusion, phage-containing hydrogels eliminate efficiently *P. aeruginosa* in wounds and may be a proper option for treating infectious wounds.

## 1. Introduction

Skin ulcers are one of the most common tissue injuries that may take a long time to heal, due for example to infections. To promote healing, the affected area must first be cleansed of infectious bacteria [1]. Due to increased antibiotic resistance, infected wounds have become more difficult to treat, and this has led to the emergence of chronic incurable wounds [2].

According to the World Health Organization (WHO), one of the most dangerous antibiotic-resistant pathogens is *Pseudomonas aeruginosa*, which poses a threat to public health. *P. aeruginosa* plays an important role in acute skin infections and chronic burns, and in the most severe cases it can cause tissue necrosis and sepsis and greatly increase the risk of death [3]. Recently, overuse of antibiotics during treatment and the presence of multiple resistance mechanisms have accelerated the development of multidrug-resistant strains and led to the ineffectiveness of conventional antibiotic therapies against these microorganisms [4,5]. In addition, due to the limited and time-consuming production of new antibiotics and the side effects of their use, alternative treatment approaches are needed. Therefore, phage therapy has gained increasing attention as an alternative treatment [6].

Bacteriophages are viruses that specifically lyse and kill only their host bacteria without affecting the body’s natural microbiome and mammalian cells [7]. Moreover, phages are very diverse and pervasive, and there are specific phages for almost every bacterium that maintain the microbial balance of the ecosystem [8]. During the treatment, phages concentrate at the site of infection [6]. They infect the host bacteria and reproduce themselves, thereby representing a self-replicating drug that is active only until the targeted host bacteria are completely eliminated [9]. The concentration of phages, unlike that of antibiotics, can increase in response to the presence of increasing concentrations of the target bacteria. Thus, a low initial phage dose may be sufficient for treatment [10]. These properties distinguish phage therapy from antibiotic treatments, and phage therapy can be an alternative to antibiotics or at least a complementary approach to treating infections [11]. To prove this, many in vitro and in vivo studies have been performed to evaluate the effectiveness of phages against various infections [12]. In such treatment experiments, the selected phages must be lytic, and preferentially a phage cocktail should be used to enable elimination of a wide range of different strains of a species. Phage cocktails are more effective than single phages and are more suitable for treating and preventing development of phage resistance in bacteria [13].

To use bacteriophages in practical applications, phages must be delivered to the site of infection in a controlled release system [10,14]. Hydrogels can be considered a potential candidate with moisture retention properties, high adsorption capacity, biocompatibility and the ability to load and release for bacteriophage delivery to heal wounds [15]. They are three-dimensional networks of hydrophilic polymer chains that have a structure similar to living tissue. In addition, hydrogels can be sensitive to pH, temperature, and enzymes, depending on the type of polymer (natural or synthetic) [16,17]. They are able to absorb wound secretions that allow the free movement of keratinocytes and fibroblasts needed for re-epithelialization. Moreover, replacing hydrogels is painless compared to other dressings because their hydrophilicity prevents cell attachment and they do not stick to the wound [17]. Hence, in recent years, several studies have been conducted on the use of hydrogels to transfer antibacterial compounds [18]. Bacteriophage hydrogels as an ideal dressing can help with various aspects of wound management, such as eliminating infection and repairing tissue in the short term [19].

In the present study, hydrogels based on two natural polymers, sodium alginate (SA) and carboxymethyl cellulose (CMC), were prepared for the delivery of phage cocktails and antibiotics to the wound site. SA and CMC are both natural polymers that are preferable to synthetic polymers in terms of abundance, non-toxicity and biodegradability. On the other hand, SA and CMC did not affect phage performance due to their lack of a antimicrobial effect, and we were able to check the real antimicrobial effect of phages. Further, in terms of antimicrobial function and skin regeneration as an ideal dressing in vitro and in vivo on wound infection caused by *Pseudomonas aeruginosa*, they were evaluated and approved in the mice model.

## 2. Materials and Methods

### 2.1. Materials

Carboxymethyl cellulose (CMC) was obtained from ACROS (Dessel, Belgium), calcium chloride (CaCl_2_) from SAMCHUN (Seoul, Korea), sodium alginate (SA) from Kia chemistry (Tehran, Iran) and sodium chloride (NaCl), magnesium sulfate six-hydrate (MgSO_4_ × 6H_2_O) and Tris from Himedia (Thane, India). Tris-hydrochloric acid (HCl), chloroform, formaldehyde and all culture media used were purchased from Merck (Darmstadt, Germany). Polyethylene glycol (PEG) was obtained from BioBASIC (Markham, Canada) and ciprofloxacin powder and antibiotic discs from Mast (Bootle, UK).

### 2.2. Determining the Pattern of Antibiotic Susceptibility

The nosocomial and coded strains of *P. aeruginosa*, stored frozen at −80 °C, were from the collection of Tabriz University. The antibiotic resistance patterns of *P. aeruginosa* were determined by MIC (micro dilution) and disc diffusion methods following the CLSI–M02 and M07 instructions [20]. The following antibiotics were tested: amikacin (30 μg), gentamicin (10 μg), tetracycline (30 μg), ciprofloxacin (5 μg), ceftazidime (30 μg), and amoxicillin (30 μg).

### 2.3. Preparation of Phages

#### 2.3.1. Isolation of Bacteriophages

For phage isolation, several water samples were collected with sterile 50 mL Falcon tubes from hospital wastewater and stagnant contaminated water in the city of Tabriz. To eliminate bacteria, 5 mL of chloroform was mixed with 25 mL of the aqueous sample and mixed thoroughly for several hours. After the chloroform settled to the bottom, the upper water phase was used to isolate phages. To enrich the phages in the sample, to 1 mL of water phase in a sterile Falcon tube, 100 μL of fresh *P. aeruginosa* ATCC 27853 culture and 8 mL of nutrient broth medium were added, and the culture was incubated at 37 °C for 18–24 h. A 1 mL aliquot of the culture was centrifuged in microtubes at 14,000 rpm for 15 min. The presence of phages in the supernatant was tested using the double-layer agar [21]. A 1% nutrient agar medium was prepared as a bottom layer. The top layer was 0.7% agar in distilled water, which was allowed to cool to 45 °C, to which 100 μL of the supernatant and 10 μL of an 8 h culture of the indicator bacteria were gently mixed before pouring over the bottom layer. After the solidification of the top layer, the plates were incubated overnight at 37 °C. The plates were inspected after 24 h incubation for the presence of phage plaques [22].

#### 2.3.2. Purification and Storage of Bacteriophage

Single plaques were removed from the plate using sterile Pasteur pipettes and transferred to tubes containing SM buffer (0.1 M NaCl, 0.016 M MgSO_4_ × 6H_2_O, 50 mL of 1 M Tris-HCl, in 1 L distilled water [pH = 7.5]). This plaque purification was repeated to ensure the purity of the isolated phages. For phage storage, phages were recovered from a semi-confluent double layer agar after incubation at 37 °C for 24 h by rinsing with 10 mL of SM buffer and collected in a sterile Falcon. The tubes were centrifuged at 6000 rpm for 15 min, and the phage-containing supernatant was separated and stored at 4 °C [22].

Two methods were used for long-time maintenance of phage stocks. To store phages at −80 °C, sterile DMSO was added (6% final concentration) to the phage suspensions. To preserve the phages by lyophilization, a solution containing 0.5 M sucrose and 1% gelatin was used. Depending on the volume of phages washed from the plate, the same amount of solution was added and after complete shaking, placed in the freezer at −20 and −80 °C overnight and then transferred to a freeze dryer.

#### 2.3.3. Bacteriophage Titration

Phage concentrations as plaque-forming units (PFU/mL) were determined by the dilution method. For this purpose, 10-fold dilution series were prepared from purified phages and 100 µL aliquots of each dilution was plated with indicator bacteria on double-layer agar. The number of plaques was counted for each dilution and used to calculate the original PFU/mL [23].

#### 2.3.4. Bacteriophage Host Range

To select suitable phages for phage therapy and phage cocktail preparation, the sensitivities of 15 different *P. aeruginosa* strains isolated from clinical samples to the isolated phages were investigated by double-layer agar culture method using 10-fold dilutions of phage stocks.

#### 2.3.5. Transmission Electron Microscopy (TEM)

For TEM, the phages were pelleted and washed once with, and resuspended into, 0.1 M neutral ammonium acetate. A 3 µL aliquot was allowed to adsorb for 30 s on a carbon-coated copper grid, and after removal of the fluid by filter paper the phages were stained using 2% uranyl acetate (pH 4–4.5). After staining, phages were observed with EM208S electron microscope at 100 kV [23].

#### 2.3.6. Phage Genome Sequencing

Phage DNAs were extracted from phage PB10 and PA19 stocks of 3 × 10^11^ and 1 × 10^11^ PFU/mL, respectively, using the Maxwell^®^ RSC Viral TNA purification kit (Cat no AS1330, Promega Corporation, Madison, WI, USA), according to the manufacturer’s protocol. The DNA samples were sequenced with the 150 bp paired-end protocol in the Illumina HiSeq platform at NovoGene (Cambridge, UK). At Novogene, the genomic DNA was randomly sheared into short fragments. The obtained fragments were end-repaired, A-tailed and further ligated with Illumina adapters. The fragments with adapters were PCR-amplified, size-selected, and purified before being subjected to sequencing. To assemble the phage genome, 100,000 reads were randomly selected from the 12,640,648 and 11,880,254 total reads and assembled using an A5-miseq integrated pipeline [24]. The presence of genome termini was evaluated using PhageTerm [25]. The assembly was verified by mapping the original reads back to the genome with Geneious Prime 2022.2.2 assembler (Biomatters Ltd., Auckland, New Zealand)

Genes and the functions of the gene products were predicted with a rapid annotation server using subsystem technology RAST [26]. The predicted genes were further confirmed and analyzed using the Artemis software(Release 17.0.1) [27]. The annotated genomic sequences of phages PB10 and PA19 were deposited to Genbank under the accession numbers OP831166 and OP831167, respectively.

### 2.4. Preparation of Hydrogel

The main natural polymer used in this study was sodium alginate (SA). To prepare the hydrogel, 4 g of SA was added slowly to 100 mL of preheated distilled water at 40 °C and stirred continuously by a magnet to obtain a homogeneous solution. Carboxymethylcellulose (CMC) polymer was prepared by adding slowly 0.2 g of CMC into 10 mL of distilled water, under constant stirring to prevent the formation of aggregates. Finally, both solutions were mixed together, and the final solution was autoclaved. After sterilization and cooling, the solution was used to prepare various hydrogels. During the hydrogel synthesis, sterile CaCl_2_ solution with different concentrations was used to crosslink the hydrogel.

### 2.5. Preparation of Bacteriophages and Ciprofloxacin Encapsulated in SA-CMC Hydrogels

Based on antibiograms and MIC tests of the *P. aeruginosa* strain, ciprofloxacin was selected as the antibiotic for these experiments. To prepare phage-, antibiotic-, or phage- plus antibiotic-containing hydrogels, 200 μL of phage cocktail, 50 μL of ciprofloxacin solution in water (10 mg/mL), or both, were added to 10 mL of the hydrogel solution and stirred thoroughly for 10 min. To convert the hydrogel into a gel, 1 mL of 1% CaCl_2_ solution was added with a syringe to the resulting hydrogel solutions. The thus obtained topical gel was stored at 4 °C. To convert the hydrogel into a film, immediately after addition of the 1% CaCl_2_ solution to the hydrogel, the resulting solution was poured onto a plate and placed at −20 °C for 18–24 h. Then 8 mL of 10% CaCl_2_ solution was poured on the hydrogel and excess liquid was drained after one hour. The obtained hydrogel films were stored at 4 °C. The final concentrations of the two selected phages, PB10 and PA19, were 3 × 10^7^ and 4 × 10^7^ PFU/mL, respectively, and that of ciprofloxacin, 50 µg/mL.

To prepare a control hydrogel, the phage and antibiotic were omitted, and only 10 mL of hydrogel base solution was used to prepare the topical gel and the hydrogel film.

### 2.6. Characterisation of the SA-CMC Hydrogels

#### 2.6.1. Determination of the Swelling Index

For this experiment, the hydrogel film was cut into 2 × 2 cm pieces and allowed to dry at room temperature. After the dry weight of the hydrogel (W_0_) was determined, it was immersed in distilled water and incubated at 37 °C. After 24 h and every day for a week, the hydrogel was weighed (W_s_). Swelling index was calculated using the formula [28].
Swelling index (%) = 100 × (W_s_ − W_0_)/W_0_

#### 2.6.2. Degradation of Hydrogels

To determine the possible degradation of the hydrogel, a piece of dry and pre-weighed 2 × 2 cm hydrogel (W0) was immersed in five ml of PBS and incubated at 37 °C. On days 3, 7 and 14, the hydrogel piece was washed three times with distilled water and weighed (Wt). The amount of hydrogel degradation on different days was calculated by the following formula [28]:Remaining weight (%) = (W_t_/W_0_) × 100

#### 2.6.3. Water Vapor Transmission Rate

The water vapor transmission rate (WVTR) of hydrogels was determined based on the ASTM E96-00 method by the US Standards Office. Two test tubes with 5 mL of distilled water were prepared. The tube opening of one of the tubes was then completely and tightly sealed with the dried hydrogel, and the other tube was left open. Both tubes were weighed (M_0_). The tubes were incubated at 37 °C and every day for a week the tubes were weighed (M_d_). The WVTR each hydrogel was calculated with the following equation [29]:
WVTR =(Δm ⁄Δt)/A


In this equation, Δm⁄Δt represents the weight of moisture lost on different days, and A represents the surface area (m^2^) of the tube opening.

#### 2.6.4. Scanning Electron Microscopy (SEM) Analysis

Scanning electron microscopy (SEM) was used to examine the surface morphology and general structure of the control and phage-containing hydrogel films. The hydrogels were completely dried and then coated with a series of thin metal coatings such as gold before observation and examination with a SEM (Tescan—15 KV) [29].

#### 2.6.5. Fourier Transform Infrared (FTIR) Spectroscopy

To confirm the presence of chemical compounds and configuration of various bonds of SA–CMC hydrogel and its constituents, including SA and CMC powder, the films were evaluated by determining the FTIR spectrum in the range of 400–4000 cm^−1^ using the TENSOR27 model FTIR device (Bruker, Germany) [19].

#### 2.6.6. In Vitro Antibacterial Activity Assay

The antibacterial potential of the SA-CMC hydrogels on *P. aeruginosa* was assessed by three different methods.

In the quantitative method, 100 μL of overnight culture of *P. aeruginosa* was inoculated into five flasks containing 50 mL of sterile saline to a final concentration 3 × 10^8^ CFU/mL [19,30]. Then, 2 × 2 cm pieces of pure hydrogel, phage, antibiotic and phage-antibiotic hydrogel were immersed in four of the bacterial suspensions, and the fifth remained as a no-treatment control. The suspensions were incubated in a shaker at 37 °C, and to evaluate the antibacterial effect of the hydrogels, the bacterial concentrations (CFU/mL) were determined from samples collected at 3, 6 and 8 h by a standard plate count on MacConkey agar.

In the qualitative method, the antimicrobial activity of hydrogels was evaluated by disc diffusion [19]. From an overnight culture of bacteria, adjusted to 0.5 McFarland, bacteria were spread with a swab on Muller Hinton agar (MHA). Discs, 6.4 mm in diameter, were cut from all four hydrogels and placed on the bacterial lawn, incubated at 37 °C for 24 h, and growth inhibition around the discs was observed.

In the agar diffusion method, the antimicrobial effect of the liquid form of the hydrogels was examined. After spreading the bacteria on MHA, four wells were punched in the agar and filled with 30 μL of the different hydrogels. The plates were incubated at 37 °C, and growth inhibition around the wells was inspected the next day.

#### 2.6.7. Stability Studies

Phage/antibiotic stability at 4 °C in SA-CMC hydrogel was investigated weekly up to 2 months. The stability of the antibacterial property of the hydrogels was determined based on the quantitative method (Section 2.6.6).

### 2.7. Animal Experiments

The animal experiments were carried out according to the instructions of the Ethics Committee of Tabriz University (Specialized Ethics Committee in Biomedical Research: IR.TABRIZU.REC.1400.055). Male BALB/c mice (*Mus Musculus*) of 25 ± 2 g in weight were purchased from the Research Center of the Tabriz Veterinary faculty and allowed to adapt to the new environment for a week before the start of the experiments. All mice were kept in single cages at 22 ± 2 °C with free access to food and water using a 12-h light/12-h dark cycle.

The mice were divided into nine groups as follows. All groups included five mice, except group B, which consisted of 10 mice.

Group without infection or control (C)

Infection without treatment (B)

Daily pure hydrogel (H)

Daily phage hydrogel (PD)

Phage hydrogel (PO)

Daily ciprofloxacin hydrogel (AD)

Ciprofloxacin hydrogel (AO)

Daily phage–ciprofloxacin hydrogel (PAD)

Phage–ciprofloxacin hydrogel (PAO).

#### 2.7.1. Wound Formation and Infection Model

To create the wound, the hairs on the back of the mice (on the lumbar spine between the vertebrae 2–6 L) were first shaved with shaving cream. The mice were anesthetized by intraperitoneal injection of ketamine (60 mg/kg). Prior to surgery, the wound site was sterilized. A wound of 5 mm in diameter was cut with a scalpel to the depth of epidermis and superficial dermis without damaging the muscles and causing minimal bleeding. The thickness of the created wound was up to the beginning of the muscle tissue (maximum skin).

The wounds (except the control group) were infected with 50 µL of a 3 × 10^8^ CFU/mL suspension of *P. aeruginosa* ATCC27853 bacteria, and after 30 min the wounds were covered with desired hydrogels or left untreated in the control group.

In groups H, PD, AD and PAD, a fresh hydrogel was applied on the wounds daily up to 14 days, while in groups PO, AO, and PAO, the hydrogel was applied only once, on the first day of the experiment, and that hydrogel remained on the wounds throughout the experiment.

#### 2.7.2. In Vivo Evaluation of Antibacterial Effect of Hydrogel

The antimicrobial efficiency of the hydrogels was monitored by sampling the wounds on days 1, 3, 7, 10 and 14 of the treatment. For sampling, each sterile swab was moistened in sterile distilled water (2 mL) prepared specifically for each wound and then was gently drawn over the wound site and placed in a tube containing 3 mL of sterile physiological saline. After thorough mixing, 100 μL aliquots were cultured on eosin-methylene-blue (EMB) agar plates. The number of colonies was counted after 24-h incubation at 37 °C.

#### 2.7.3. Wound Healing

To evaluate the healing process, the wounds were photographed and their diameters in mm were measured with a caliper on days 1, 3, 7, 10 and 14. The wound size reduction was calculated by the following equation:Wound contraction (%) = (A_0_ − A_T_)/A_0_ × 100
where A_0_ shows the area of the primary wound and A_T_, the area of the wound on different days of treatment.

#### 2.7.4. Blood Cultures of Dead Mice

Some mice died during the wound infection experiment. To find out whether the death was caused by *P. aeruginosa*, a blood sample (ca. 200 µL) was taken directly from the heart using a sterile syringe, inoculated into 25 mL of TSB and incubated at 37 °C for 18–24 h. Then, 100 μL of TSB culture was spread on nutrient agar, MacConkey and EMB plates. Identification of *P. aeruginosa* growth confirmed that the cause of the death was bacteremia/sepsis caused by the test bacteria.

#### 2.7.5. Histopathological Studies

Skin samples of the wounds were taken at the end of the 14-day experiment and fixed at room temperature for 72 h in 10% formalin. The tissue blocks were first dehydrated in an alcohol gradient in special cassettes, after which they were immersed in xylene for one hour. The tissues were then frozen in paraffin molds and cut with a microtome to 5–10 µm slices and mounted on slides containing gelatin. For staining, the slides were immersed twice in xylene and then hydrated in an alcoholic gradient and washed in water. The slides were stained with hematoxylin and eosin. After dehydrating the stained samples with alcohol gradient and clarification with xylene, they were washed and dried. A drop of Permount was added to the tissue of each slide and covered with a coverslip [31]. Finally, the slides were examined for tissue changes under a microscope.

#### 2.7.6. Statistical Analysis

Statistical analysis of data was performed using the Mini-tab v19 software. The experiments were repeated several times on different days. Data were expressed in terms of means. Statistical analysis of data related to microbial load and wound size in different groups of mice (within each group and the corresponding days in different groups) were analyzed by one-way ANOVA with Bonferroni and Tukey post-tests.

## 3. Results and Discussion

### 3.1. Antibiotic Susceptibility Pattern

Disc diffusion and MIC methods were used to determine the antibiotic resistance of *P. aeruginosa* ATCC27853 bacteria. According to the CLSI standard table and the results obtained from both methods, the *P. aeruginosa* strain was sensitive to ciprofloxacin and amikacin, moderately resistant to gentamicin and tetracycline, and resistant to ceftazidime and amoxicillin. Based on the results, ciprofloxacin was selected as the preferred antibiotic for the treatment of *P. aeruginosa* infection in this project.

### 3.2. Characterization of Bacteriophages

Today, using rapid and modern sequencing techniques and a better understanding of bacteriophage genetics, phage therapy is a promising new method [8,9]. Many in vitro and in vivo studies have been performed to evaluate the efficacy of phages against *P. aeruginosa* infections [32]. For example, in a 2015 study, six bacteriophages, DL52, DL54, DL60, DL62, DL64, and DL68, were isolated with a range of effects and used as phage cocktails against *Pseudomonas* PAO1 infection [33]. In the present study, 10 specific phages with plaques of different appearances, some clear and some opaque, were isolated from the sewage against *P. aeruginosa*. Phages were abbreviated based on the first letter of *Pseudomonas*, sampling series and collected sample number. After isolation and purification, the concentration of each phage was determined. Based on phage titration, it was found that there was an average of about 10^7^ PFU/mL of phage in each suspension. By phage typing test, the effect of each of 10 phages on 15 different clinical antibiotic-resistant *P. aeruginosa* isolates was investigated (Table 1). All the isolated phages were able to infect several of the *P. aeruginosa* isolates. The phage PB10 had the widest host range with 11 strains, followed by phages PA19 and PA13 with nine strains. Based on these results, phages PB10 and PA19 were selected for a phage cocktail for preparation of the hydrogels for phage therapy on experimental wound infections in mice. These phages provided the widest host range and had completely clear plaques, predicting a strictly lytic nature for the phages (Figure 1). Both phages had long contractile tails and icosahedral capsids of ca. 70 nm in diameter in TEM (Figure 2), suggesting that morphologically they represent myoviruses [34].

### 3.3. Phage Genome Characterization

The sequences of the genomic DNA of phages PB10 and PA19 were determined using the Illumina paired-end technology. The assembled genomes were 66,096 and 65,660 bp in size, respectively. PhageTerm analysis predicted that both phages use headful packaging starting from a pac site. Comparison of the genomes demonstrated that the phages were closely related to each other with 86% identical collinear genomes (Figure 3A). Comparison of the genomes to known phages revealed that they are >90% identical to a growing group of PB1-like *Pseudomonas* phages belonging taxonomically to the genus *Pbunavirus* of the *Caudoviricetes* class [35,36].

Comparison of the putative receptor-binding protein sequences encoded by the gene g62 of both phages revealed distinct differences in the C-terminal third of the proteins (Figure 3B), which can explain the host range differences between the phages (Table 1).

### 3.4. Preparation and Characterization of Hydrogels

In phage therapy applications, phages should be delivered to the site of infection in a controlled way [37]. Biocompatible hydrogels have lately been widely used as wound dressings due to their optimal properties, which include high hydration, a soft and non-sticky constitution, and easy and painless replacement [38]. A hydrogel-based delivery system loaded with bacteriophages was used successfully to prevent binding and colonization of multidrug-resistant *Enterococcus faecalis* around and inside femoral tissues, a promising way to avoid bacterial contamination during bone grafting [15]. Hydrogels composed of two different polymers have improved biological stability and better physical properties than single polymer-hydrogels and are therefore better suited as wound dressings [16,39]. The hydrogels used in this work contained natural polymers 4% SA and 2% CMC. Cross-linking was carried out with 5% and 10% CaCl_2_ in both film and gel modes with antimicrobial agents including cocktails. Either phage or ciprofloxacin or both were loaded during hydrogel preparation to obtain three differently loaded hydrogels, i.e., phage-, ciprofloxacin- and phage-ciprofloxacin-hydrogels. Pure hydrogel was used as a control (Figure 4).

#### 3.4.1. Swelling Index Analysis

The SA-CMC hydrogel showed swelling of 15–20% after day 3 when compared to the day 1 situation, being rather stable until day 7 (Figure 5A). Swelling of the hydrogel increases the release of antimicrobials. Excessive swelling should be avoided, as it may cause hydrogel instability, making moderately swelling hydrogels more suitable for wound healing [40,41].

#### 3.4.2. In Vitro Degradation Behavior of Dressing

Degradation of the hydrogels was followed over 14 days, during which the SA-CMC hydrogel degraded such that the hydrogel lost almost 60% of its weight. The decomposition of the dressings has been attributed to the degradable properties of polysaccharides, which has been considered important for wound healing [29]. Along with hydrogel degradation, it can be anticipated that the release of antimicrobial agents from the dressing increases (Figure 5B).

#### 3.4.3. Water Vapor Transfer Rate of Hydrogels

The water vapor transfer rate (WVTR) describes the important ability of the hydrogel to resist drying. During the 7-day experiment, 30% less water evaporated from the tubes coated with hydrogel than from the control tubes (Figure 6), and the observed WVTR values remained within the range of 1500–2000 g/m^2^ in 24 h, which has been shown to maintain good moisture [29]. Therefore, hydrogels can provide a suitable environment for the proliferation of epidermal cells and fibroblasts [42].

#### 3.4.4. FTIR Analysis

The FTIR spectra of SA, CMC and SA-CMC hydrogels were determined (Figure 7) and compared and confirmed with other recent studies [43]. In both SA and CMC diagrams, the peaks in the range 3400–3500 cm^−1^ are related to the hydroxyl bond (O-H), while the peaks in the range 2900–3000 cm^−1^ are related to the C-H bond. In the SA diagram, the peaks at 1427 cm^−1^ to 1639 cm^−1^ are related to the C=O bond, the 1125 cm^−1^ peak is related to the C-O-C bond, and the 615 cm^−1^ peak is related to the O-H bond. In the CMC diagram, the peak at 1624 cm^−1^ is related to C=O bond, the peak at 1425 cm^−1^ is related to CH_2_ bond, the peak at 1063 cm^−1^ is related to CH-O-CH_2_ and the peaks in the range of 600–700 cm^−1^ and 1329 cm^−1^ are related to O-H bond. In the SA-CMC hydrogel diagram, the peak at 3418 cm^−1^ and 580 cm^−1^ corresponds to the OH bond and is higher than at spectra SA and CMC; the peak at 1637 cm^−1^ indicates the C=O bond, the peak at 1459 cm^−1^ indicates the CH_2_ bond, and at 1162 cm^−1^ indicates the C-O-C bond [43,44]. In this analysis, all three spectra were compared, and based on the presence of similar peaks at the desired wavelengths in all three graphs, the presence of SA and CMC in the hydrogel was confirmed.

#### 3.4.5. Hydrogel Surface Morphology

The surface morphology of the pure hydrogel and the phage-containing hydrogel were examined by SEM with a voltage of 15 kV. As shown in Figure 8, the surface of the pure hydrogel was smooth, while small protrusions were observed on the surface of the phage-containing hydrogel, which is probably due to the presence of phage and phage solvent ingredients (SM buffer).

### 3.5. In Vitro Antibacterial Activity Assays

In the two disc diffusion experiments, i.e., diffusion from a hydrogel disc and from a hydrogel-filled well, the phage- and ciprofloxacin-containing hydrogels inhibited the growth of bacteria (Figure 9). However, the inhibition halo around phage-antibiotic hydrogels was larger than those of the others, demonstrating a synergistic effect between them. Pure hydrogel served as a negative control without antibacterial properties.

In the study of the antibacterial effect of various hydrogels by the quantitative methods, analysis was performed at 3, 6 and 8 h. This was reflected in the reduction of bacterial numbers (Figure 10). At 8 h, the phage-antibiotic hydrogels (95% reduction) were more efficient in killing than the phage-containing (86%) and antibiotic-containing (88%) hydrogels, again indicating a synergistic effect of phages and antibiotics. These findings are consistent with the synergistic effect noticed between phage PEV20 and ciprofloxacin against *P. aeruginosa* [45].

While the number of bacteria increased in the control and pure hydrogel groups over time, bacteria were killed in the phage-, antibiotic-, and phage-antibiotic-hydrogels over time (Figure 10). These results were corroborated by the qualitative assays (Figure 9).

### 3.6. Stability of the Antibacterial Efficiency of the Hydrogels

The killing efficiency of the antibacterial hydrogels was assessed over time (Figure 11). All three hydrogels were able to maintain their antibacterial activity up to 8 weeks, although some activity loss was evident already at 2 weeks and more clearly at 4 and 8 weeks, when the activity was estimated to be 52% (Figure 11). These results are comparable or better to those reported by others [19].

### 3.7. Animal Experiments

#### 3.7.1. In Vivo Antibacterial Activity

In the animal experiment, mice were divided into nine treatment groups. Bacterial numbers in the wounds were assessed over time using swab cultures (Figure 12). The bacterial loads in the untreated (B) and pure hydrogel (H) groups were highest on day 1, after which the numbers gradually decreased until day 12. In all the groups that were treated with antibacterial hydrogels, the bacterial numbers were very low already on day 1, and the wounds were practically sterile on day 3. Wounds treated with phage-antibiotic hydrogel (PAD and PAO) were almost sterile on the first day and performed better than PD, PO or AD, AO groups, indicating a synergistic effect between phage and antibiotics (Figure 12). While all the mice treated with antibacterial hydrogels survived and were in good general condition, 60% of group B and 45% of group H mice died. Blood cultures of the dead mice were positive for *P. aeruginosa*, indicating that the mice died of sepsis caused by the bacteria.

#### 3.7.2. Wound Healing

In this study, a hydrogel containing a phage cocktail and ciprofloxacin was prepared with a combined antimicrobial function and wound regeneration. The progress of wound healing and epithelialization were evaluated in the different mouse groups (Figure 13). The wound-healing process was best in the PD and PAD mouse groups where the healing started already on day 1, and the wounds were completely healed at day 14, in contrast to the other treatment groups. The healing process in the AD and AO groups started with a delay, at days 3 and 7, respectively. The healing was only moderate in the C and H mouse groups. Overall, the healing was slowest in the groups B and AO.

The steepest slopes in the wound-size curves (Figure 14) for phage- and phage-antibiotic hydrogels (PD, PO, PAD and PAO) also illustrate the high rapidity of wound healing in these groups.

These results are comparable to those with a hydrogel containing bacteriophage T7 and an acidic fibroblast growth factor against *E. coli* infection where the wound was sterile on day 7 and the wound healing started on day 3 [18]. Similarly, in an experiment where phage MR10 and minocycline containing hydrogel was used to treat an MRSA infection, the wounds became bacteria-free on day 3; however, the wound-healing process was somewhat slower then reported here [19]. In conclusion, phage-hydrogels are able to enhance skin regeneration and cure bacterial wound infections.

#### 3.7.3. Histopathological Studies

Histological examinations of tissue samples recovered on day 14 from the wounds revealed that in group B and H mice the epidermis had not yet healed, showing both a crack and visible wound secretion, and there were signs of severe inflammation and fibrosis (Figure 15). Additionally, slight hyperkeratosis and moderate fibrosis were seen in group H. The group C wounds were non-inflammatory and almost healed as they were not infected with the bacteria. Importantly, in the PD, AD and PAD groups, no fibrosis and inflammation were observed, and the skin layers were completely regenerated (Figure 15). It is worth noting that in the PD and PAD groups, the healing process and epithelialization were faster based on the finding that the hair follicles, glands and blood vessels were quite similar to those of normal skin.

These histological findings are comparable to those in similar studies [18,19]. Therefore, SA-CMC hydrogels loaded with phage cocktails have an antibacterial effect, are able to enhance skin repair in a short time and can be offered as an ideal dressing.

## 4. Conclusions

In the present study, natural polymer hydrogels of SA and CMC loaded with a phage cocktail and/or ciprofloxacin were prepared. The hydrogels were shown to possess ideal dressing properties, including a suitable swelling index, ability to absorb secretion, moisture retention, biocompatibility and loading potential of the material and antibacterial properties. Both in vitro and in vivo experiments carried out with *P. aeruginosa* showed that the phage cocktail hydrogel used as a wound dressing reduced significantly the bacterial load of infected wounds and allowed rapid wound healing without any side effects. In addition, phages combined with ciprofloxacin had a clear synergistic effect and accelerated wound contraction and healing. Our results support further that phage therapy should be used to treat infected wounds as an alternative treatment or as a supplement to antibiotics.

## Figures and Tables

**Figure 1 viruses-15-00803-f001:**
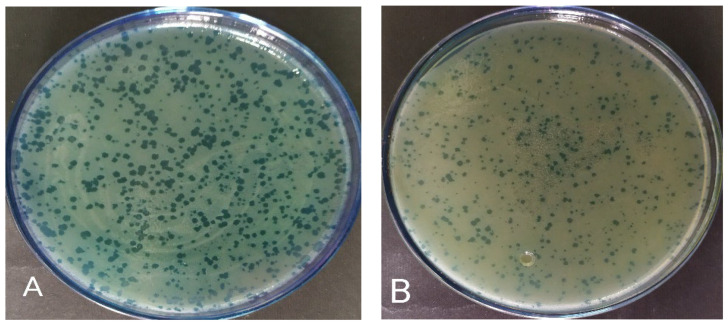
Plaques of phages (**A**) PB10 (10^7^ PFU/mL) and (**B**) PA19 (10^7^ PFU/mL) on *P. aeruginosa* ATCC27853.

**Figure 2 viruses-15-00803-f002:**
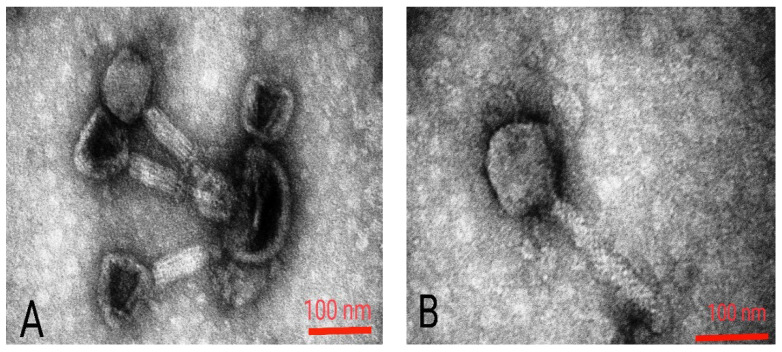
Morphology of phage particles under transmission electron microscope. Phage PB10 (**A**) and PA19 (**B**). The scale bars represent 100 nm.

**Figure 3 viruses-15-00803-f003:**
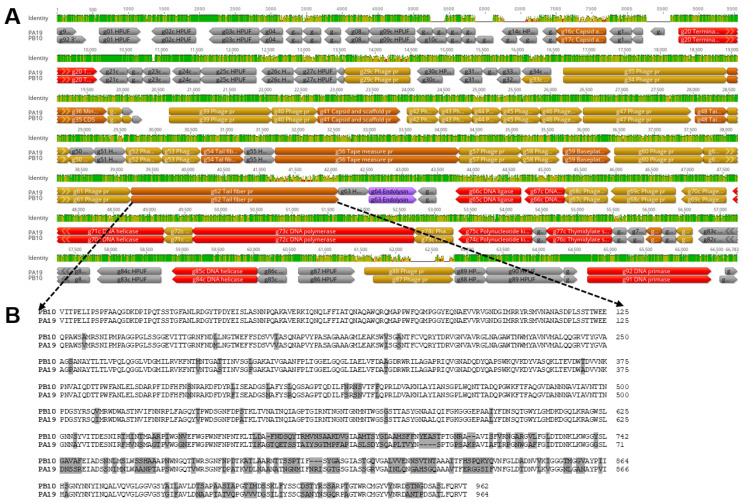
Genome comparison of phages PB10 and PA19. Panel (**A**). Alignment of the genomes. The DNA-identity-% is indicated by the “Identity” track with 100% identity, indicated by green color, while the height of the graph indicates the identity%. The alignment was generated by the Geneious version 2022.2.2 using the Clustal Omega vs. 1.2.2. Panel (**B**). Alignment of amino acid sequences of the Gp62 tail fiber proteins demonstrating the lower similarity between the C-terminal thirds of the proteins; the differences are highlighted in grey.

**Figure 4 viruses-15-00803-f004:**
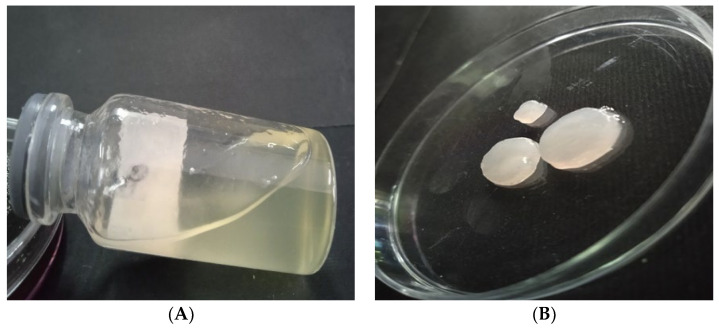
(**A**) Hydrogel in the form of gel, (**B**) Hydrogel film form.

**Figure 5 viruses-15-00803-f005:**
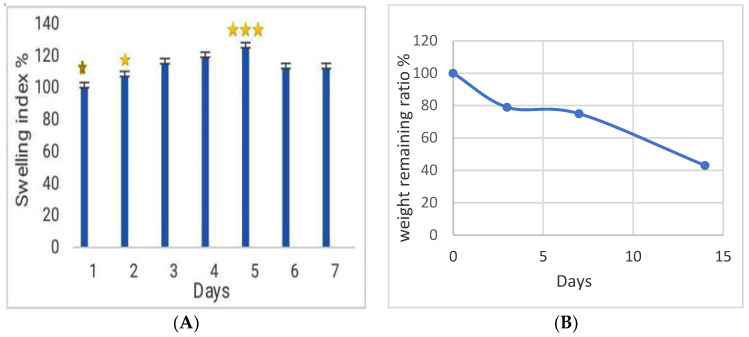
(**A**) The swelling index and 
(**B**) degradation of SA-CMC hydrogels. The results are based on three repeats showing the average with standard deviations. 
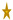
 Based on the analysis of Figure 5A, hydrogel swelling on day 5 is significant with the rest of the days (*p* < 0.001) and the day 1 is significant with the rest of the days, except for day 2, at the level of *p* < 0.05. Moreover, day 2 is significant with day 4 (*p* < 0.05).

**Figure 6 viruses-15-00803-f006:**
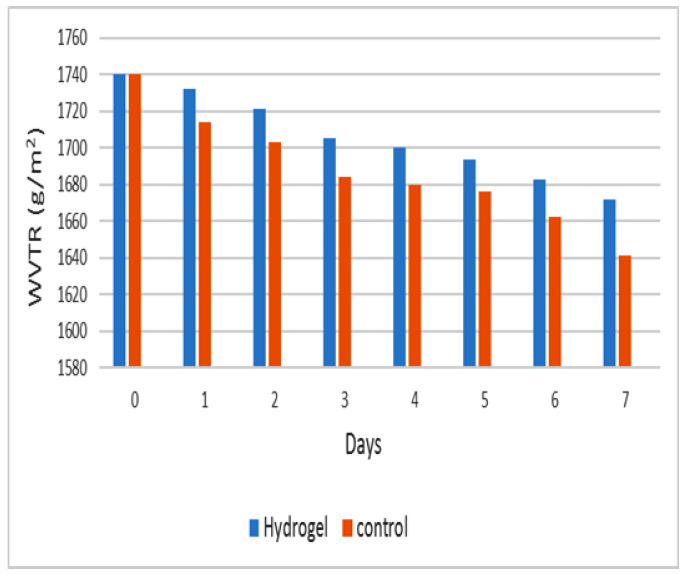
Water vapor transfer rate of hydrogel.

**Figure 7 viruses-15-00803-f007:**
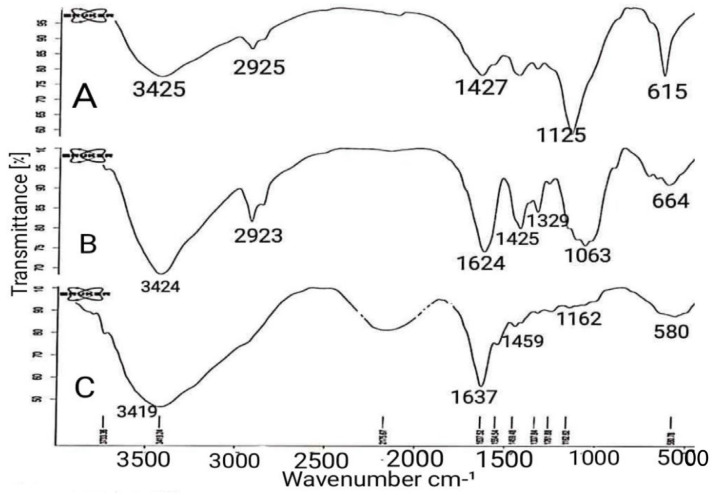
FTIR spectra of (**A**). SA, (**B**). CMC, (**C**). Hydrogel. The peaks in the range 3400–3500 cm^−1^ are related to (O-H), those in the 2900–3000 cm^−1^ range to (C-H) bonds, those in the 1427–1639 cm^−1^ range to (C=O) bonds, and at 1125 cm^−1^ to (C-O-C) bonds.

**Figure 8 viruses-15-00803-f008:**
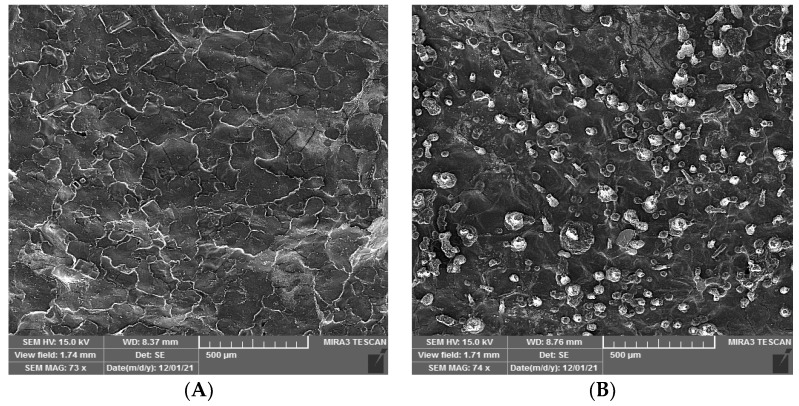
Surface morphology of (**A**). Pure hydrogel with uniform and smooth surface, (**B**). Phage-containing hydrogel with protrusion surface.

**Figure 9 viruses-15-00803-f009:**
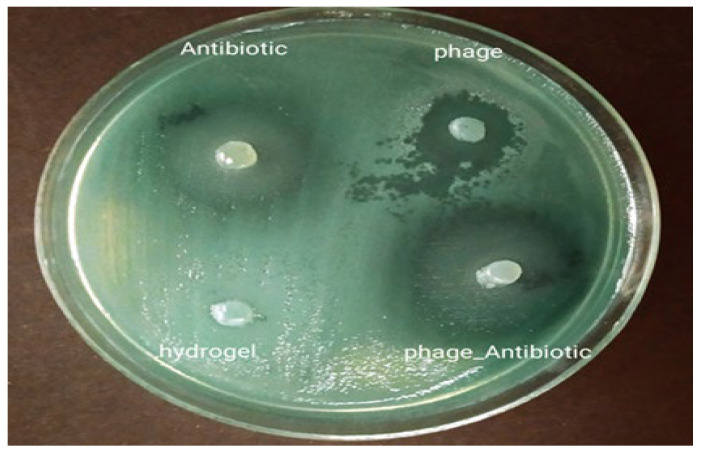
Inhibition zones of the hydrogels against *P. aeruginosa*.

**Figure 10 viruses-15-00803-f010:**
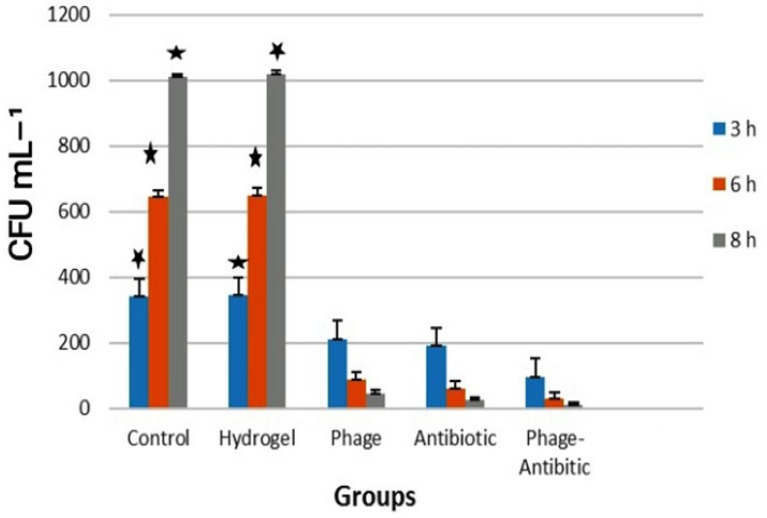
Quantitative antibacterial assay. The colony count results of different groups. The results are based on four repeats showing the average with standard deviations. 
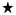
 The statistical analysis of the five groups in all the tested hours indicated a significant difference between the two control and hydrogel groups with the other three groups at the level of *p* < 0.01.

**Figure 11 viruses-15-00803-f011:**
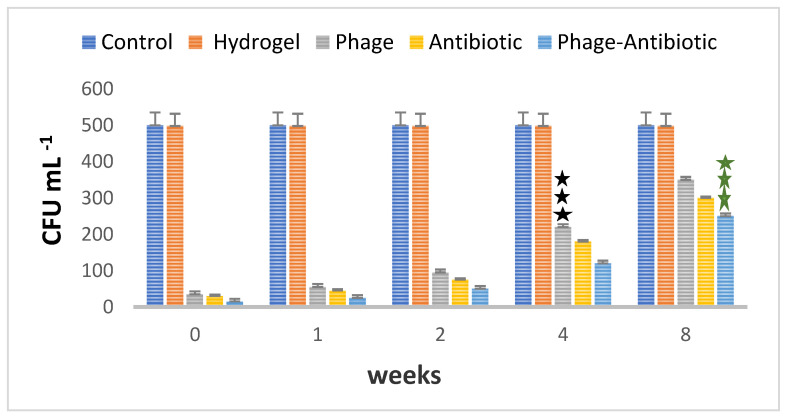
Evaluation of the stability of the phages in the hydrogels over 8 weeks, as estimated by the reduced killing efficiency over time. The results are based on three repeats, showing the average with standard deviations. 
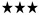
 Based on the analysis, it was found that phage 4 is significant with all other weeks of the phage group (*p* < 0.001). 
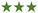
 indicates that the phage–antibiotic 8 is significant with the other weeks of the phage–antibiotic group at the *p* < 0.001 level.

**Figure 12 viruses-15-00803-f012:**
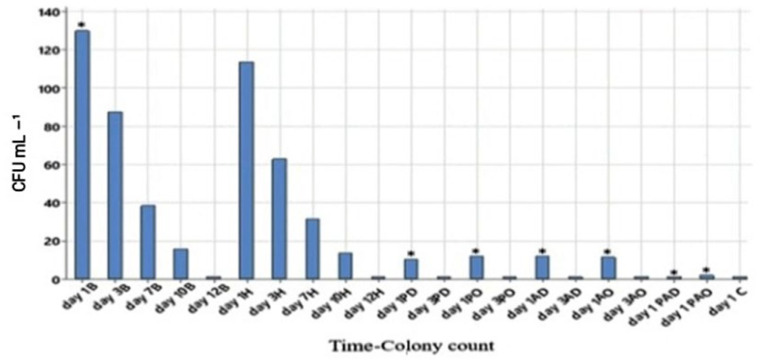
Bacterial load in different mouse groups on days 1, 3, 7, 10 and 12. C: Control, B: Untreated infectious, H: Hydrogel, PD: Phage (daily), PO: Phage (once), AD: Antibiotic (daily), AO: Antibiotic (once), PAD: Phage-Antibiotic (daily), PAO: Phage-Antibiotic (once). * *p* < 0.001 means significant relative to untreated infectious group.

**Figure 13 viruses-15-00803-f013:**
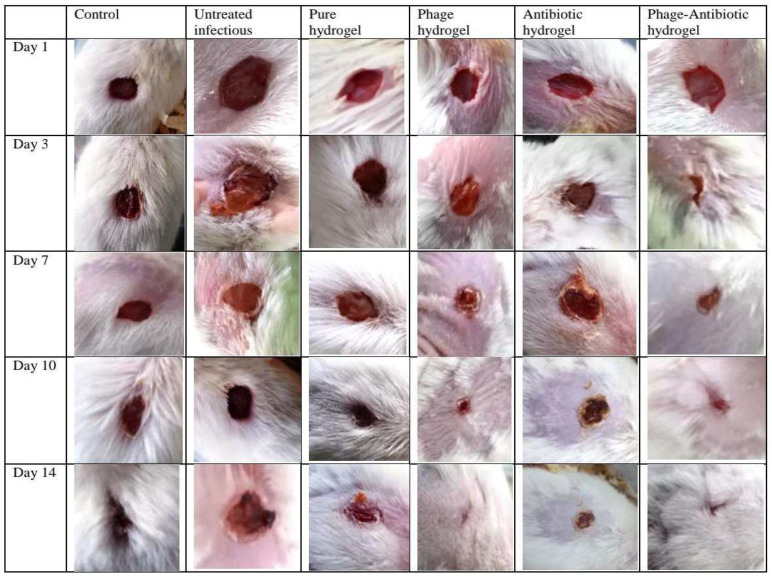
Pictures of wound-healing process over 14 days in different groups.

**Figure 14 viruses-15-00803-f014:**
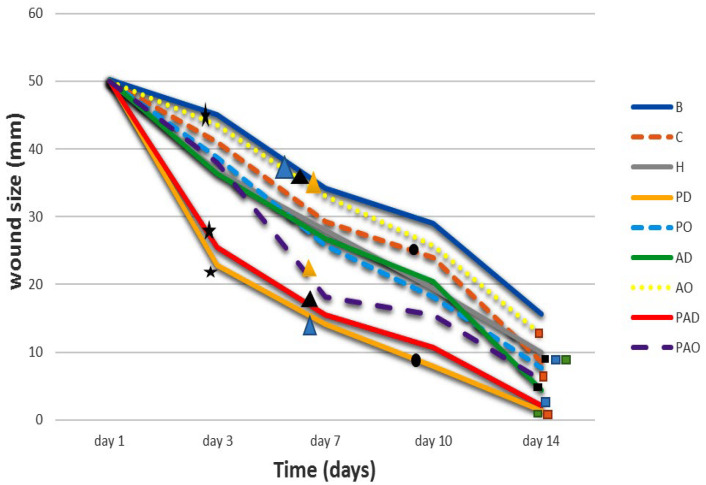
The wound healing rates based on wound sizes in different mouse groups. C: Control, B: Untreated infected wounds, H: Hydrogel, PD: Phage (daily), PO: Phage (once), AD: Antibiotic (daily), AO: Antibiotic (once), PAD: Phage-Antibiotic (daily), PAO: Phage-Antibiotic (once). According to the figure, the wound sizes of all the groups were not significantly different from each other on day one. 
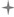
 indicates that the wound size of group AO was significantly different from those of groups PD and PAD on day 3 of the treatment (*p* < 0.05). 
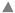
 indicates that on day 7 of the treatment, the wound size of the AO group was significantly different to those of the PD, PAD and PAO groups (*p* < 0.01). 
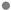
 indicates that the wound sizes of groups C and PD were significantly different on day 10 of the treatment (*p* < 0.01). 
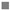
 Based on the analysis, the wound size of the group C was significantly different from those of groups AD, PAD and PD, and the wound size of group AO was significantly different from those of groups PD and PAO on day 14 of the treatment (*p* < 0.01).

**Figure 15 viruses-15-00803-f015:**
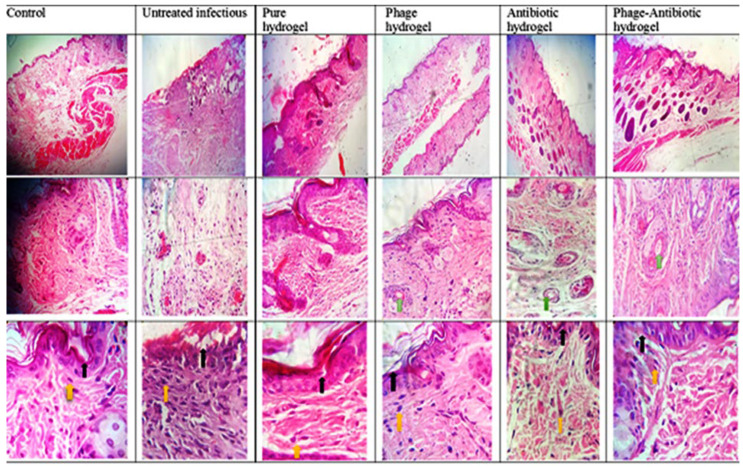
Histopathological sections of the skin in different groups. The black arrows show the repaired epidermis, the yellow arrows the macrophages and the green arrows the hair follicles.

**Table 1 viruses-15-00803-t001:** Phage typing results on *Pseudomonas aeruginosa*.

Ps15	Ps14	Ps13	Ps12	Ps11	Ps10	Ps9	Ps8	Ps7	Ps6	Ps5	Ps4	Ps3	Ps2	Ps1	
-	(+)	-	-	++	-	-	(+)	++	(+)	(+)	+	-	-	+	PA25
-	+	-	-	++	+	+	+	+	++	++	+	-	-	-	PA19
-	+	-	-	++	-	(+)	-	-	++	++	-	-	-	+	PB9
+	+	-	++	++	-	*-*	-	+	++	++	+	+	++	+	PB10
-	+	-	-	++	(+)	(+)	-	+	++	++	+	-	-	(+)	PA24
-	(+)	-	-	++	-	++	*-*	(++)	(++)	*++*	(+)	+	-	+	PA13
-	(+)	-	-	++	-	+	-	++	-	++	+	-	-	-	PC8
-	-	-	-	++	-	+	-	++	-	++	+	-	-	-	PC5
-	+	-	-	++	-	+	-	-	++	++	(+)	*+*	-	(+)	PC4
-	-	-	-	++	-	(+)	-	++	-	++	+	-	-	+	PC2

+: the number of plaques is low or intermediate, ++: the number of plaques is high, ( ): the plaques are cloudy. Ps1–Ps15: 15 clinical antibiotic resistant *P. aeruginosa* isolates.

## Data Availability

Data is available on request.

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
