# Peer review of "Design of Phage-Cocktail–Containing Hydrogel for the Treatment of Pseudomonas aeruginosa–Infected Wounds"

_viruses, 2023, doi:10.3390/v15030803_

Round 1

Reviewer 1 Report

1. These elements (sodium alginate, carboxymethyl cellulose) added by the authors in building hydrogels and their respective advantages should be explained. In this way, the reader will have a clearer idea of the features of this article.

2. Background descriptions for hydrogel dressings can be strengthened by citing 10.1021/acsami.1c25014; 10.1016/j.cej.2022.135691 and what are the advantages of the current work compared to published articles?

3. There are some formatting errors in the article. For example, space is required between the number and the character (such as Line 228). Kindly check for correctness.

4. Figure 13, does the gel achieve a long-lasting antimicrobial effect (8 weeks), does the gel degrade completely in the dressing application or does it remain in the wound?

5. Where are these hydrogels going to be used in real life? Advantages of the designed wound dressings can be improved by comparing and citing 10.1016/j.ijbiomac.2021.12.007; 10.1021/acsmacrolett.2c00290. Is phage-cocktail therapy clinically highly controversial?

6. Does the efficacy of Pseudomonas aeruginosa killing mainly originate from the phage-cocktail or the hydrogel matrix?

Reviewer 2 Report

I started reading the article “Design of phage-cocktail –containing hydrogel for the treatment of Pseudomonas aeruginosa –infected wounds” with great enthusiasm and expectations, unfortunately the article has serious shortcomings.
First, the authors did not indicate which strains of P. aeruginosa they used, although not all strains in the Tabriz University collection are susceptible to cocktail phages (Table 1). In the methods section, there is only a cursory mention of strains from the university collection, and the section "bacterial strains" itself is missing. This makes the work irreproducible and greatly reduces reader interest in the article.
Second, despite the fact that the genome sequencing section is presented in the methods section, phage genomes are not presented and no reference is made to depositing them in the genome database. In fact, the work deals with phages, about which little is known other than their name.
There is a lot of experimental material in the work regarding the hydrogel itself, but the design of experiments with phages raises serious questions.
Figure 9 shows an experiment of the combined action of an antibiotic and a phage, however, the size of the zones of inhibition (diffusion of phages and antibiotic) is surprising. The photo was taken poorly, and insufficient to confirm the conclusions of the authors. There is no statistical analysis of zones of inhibition, which leads to the conclusion that the experiment was done only once or only once it turned out with the desired result.
Figure 13 also raises questions. Different wound sizes cause differences in healing. The size of the wound either decreases or increases, which is apparently the result of either errors in the arrangement of pictures or incorrect photography of the wound. In any case, Figure 13 cannot serve as an illustration of the success of the hydrogel. 
In fact, the article should be completely revised, the necessary data and statistical analysis added, and new figures made in order to be of interest to readers.

Reviewer 3 Report

This is a comprehensive work on wound treatment using phages including phage isolation and purification, characterisation and in vitro and in vivo antibacterial testing.  The quality of the paper will be improved by having the following points considered and addressed.

Line 252: What is the species of the mice used?

Lines 272-273: How do you determine the the depth of epidermis and superficial dermis to ensure correct location and reproducibility of the cut for wound formation?

Line 275: Treatment started only 30 minutes after inoculation with bacteria. It’s very likely the bacteria still resided mostly on the wound surface and not penetrated inside tissues.  Thus, it is not surprising that the treatment worked well.  However, what is surprising is the antibiotic treatment did not work as well as the phage only treatment.   An explanation would be necessary.

Lines 277-280: how did the authors ensure the hydrogel stayed on the wound and not ran off as the animals moved or groomed?

Lines 283-284: provide missing details on how the wet sterile swab was applied to the wound site?

Line 286: The number of colonies counted after 24-h incubation reflected not just bacterial killing by phages at the wound site, but also in vitro killing during incubation.  How was the latter accounted for?

Round 2

Reviewer 2 Report

The revised version of the article looks much better. The indication of strains and references to phages give the work a scientific meaning. However, this is not enough.

After clarifying the objects of study, it became clear that the choice of phages for the cocktail was not obvious from the article. It is clear that a cocktail of phages is better than individual phages, but we do not see studies comparing individual phages across the scope of the study. This would make some sense.

Now the conclusions of the article are obvious - the phage cocktail is effective against the P. aeruginosa strain from which it was isolated. The hydrogel also improves healing, so the combination of the hydrogel with the phage cocktail leads to the expected results.

I can't buy the effectiveness of the combination of phages and antibiotics. This is simply not supported by experiments, so it is not correct to mention this as a result. Figure 13 clearly confirms this. The phage cocktail works effectively both on its own and with an antibiotic. And it's even a little better on its own. The data in Figure 9 indicate that the zone of inhibition of the antibiotic phage and the antibiotic together with the phage cocktail is also the same. We do not see any effect of PAS, so phage-antibiotic synergy is not observed in experiments.

Well, the captions to the drawings leave much to be desired. Figure 14 is not clear, low quality. The y-axis label has no dimension of values. No statistics available. If one animal was used for each experiment (which is contrary to paragraph 2.7), then this is not acceptable, and if several, then statistics are simply required. I do not believe that there were no discrepancies in the experiments.

In figure 2, the scale bar needs to be increased - it is very hard to read.

The caption in figure 1 should indicate the dilution (titer) of phages.

General note for figures: for all graphs where several experiments were done, please add statistics.

Author Response

Response to Reviewer:

Thank you for your review of our paper. We have answered each of your points below.

The revised version of the article looks much better. The indication of strains and references to phages give the work a scientific meaning. However, this is not enough.

After clarifying the objects of study, it became clear that the choice of phages for the cocktail was not obvious from the article. It is clear that a cocktail of phages is better than individual phages, but we do not see studies comparing individual phages across the scope of the study. This would make some sense.

Reply: In this study we choice phages for the cocktail based on the host range of the phages and want to prepare a cocktail with the wide host range. According to the result in table 1, phage PB10 is active against all clinical Pseudomonas isolates except P8-10 and P13; the phage PA19 is active against P8-10. So, we select the phage PB10 and PA19 for phage cocktail.

Now the conclusions of the article are obvious - the phage cocktail is effective against the P. aeruginosa strain from which it was isolated. The hydrogel also improves healing, so the combination of the hydrogel with the phage cocktail leads to the expected results.

Reply: Thank you so much.

I can't buy the effectiveness of the combination of phages and antibiotics. This is simply not supported by experiments, so it is not correct to mention this as a result. Figure 13 clearly confirms this. The phage cocktail works effectively both on its own and with an antibiotic. And it's even a little better on its own. The data in Figure 9 indicate that the zone of inhibition of the antibiotic phage and the antibiotic together with the phage cocktail is also the same. We do not see any effect of PAS, so phage-antibiotic synergy is not observed in experiments.

 Reply: We performed the qualitative test or the same as Figure 9 several times and got the same results that the measurement of halos showed the synergistic effect of phage and antibiotic. In addition, the synergistic effect of phage and antibiotic was observed quantitatively (Figure 10) in terms of colony reduction in laboratory conditions and also in vivo in terms of bacterial destruction in mouse wounds.

Also, in most phage articles (J Biomed Sci 2022; 29:23, doi.org/10.1186/s12929‑022‑00806‑1), we see the combined use of antibiotics and phage as a control group along with phage alone. This is probably because the medical system still does not believe in phage as a new treatment approach, and we should also include antibiotics in treatment trials.

Well, the captions to the drawings leave much to be desired. Figure 14 is not clear, low quality. The y-axis label has no dimension of values. No statistics available. If one animal was used for each experiment (which is contrary to paragraph 2.7), then this is not acceptable, and if several, then statistics are simply required. I do not believe that there were no discrepancies in the experiments.

Reply: Figure 14 was modified. As mentioned in Section 2.7, mice were tested in 9 groups, all groups included 5 mice except group B which included 10 mice. Also, one-way ANOVA analysis has been performed for this test.

In figure 2, the scale bar needs to be increased - it is very hard to read.

Reply: Now corrected.

The caption in figure 1 should indicate the dilution (titer) of phages.

Reply: Now it is added to the caption.  

General note for figures: for all graphs where several experiments were done, please add statistics.

The tests related to figures 5a, 10 ,11 and 14 have been repeated several times and analysis has been done for them and only significant results have been recorded. If needed, statistical analysis can be attached in the supplementary data.

Figure 5b: The graph drawn is only to show the weight loss ratio of the hydrogel during different days, which is without statistical analysis.

Figure 6: This test was done once (without repetition) and it was done only to show the moisture retention power of the hydrogel.
